# Programmed Cell Death Alterations Mediated by Synthetic Indole Chalcone Resulted in Cell Cycle Arrest, DNA Damage, Apoptosis and Signaling Pathway Modulations in Breast Cancer Model

**DOI:** 10.3390/pharmaceutics14030503

**Published:** 2022-02-24

**Authors:** Radka Michalkova, Martin Kello, Zuzana Kudlickova, Maria Gazdova, Ladislav Mirossay, Gabriela Mojzisova, Jan Mojzis

**Affiliations:** 1Department of Pharmacology, Faculty of Medicine, Pavol Jozef Šafárik University, 040 01 Kosice, Slovakia; radka.michalkova@student.upjs.sk (R.M.); maria.gazdova@student.upjs.sk (M.G.); ladislav.mirossay@upjs.sk (L.M.); 2NMR Laboratory, Institute of Chemistry, Faculty of Science, Pavol Jozef Šafárik University, 040 01 Kosice, Slovakia; zuzana.kudlickova@upjs.sk; 3Department of Experimental Medicine, Faculty of Medicine, Pavol Jozef Šafárik University, 040 01 Kosice, Slovakia; gabriela.mojzisova@upjs.sk

**Keywords:** breast cancer, chalcones, antiproliferative, apoptosis, autophagy

## Abstract

Although new chemotherapy significantly increased the survival of breast cancer (BC) patients, the use of these drugs is often associated with serious toxicity. The discovery of novel anticancer agents for BC therapy is expected. This study was conducted to explore the antiproliferative effect of newly synthesized indole chalcone derivative ZK-CH-11d on human BC cell lines. MTT screening, flow cytometry, Western blot, and fluorescence microscopy were used to evaluate the mode of cell death. ZK-CH-11d significantly suppressed the proliferation of BC cells with minimal effect against non-cancer cells. This effect was associated with cell cycle arrest at the G2/M phase and apoptosis induction. Apoptosis was associated with cytochrome *c* release, increased activity of caspase 3 and caspase 7, PARP cleavage, reduced mitochondrial membrane potential, and activation of the DNA damage response system. Furthermore, our study demonstrated that ZK-CH-11d increased the AMPK phosphorylation with simultaneous inhibition of the PI3K/Akt/mTOR pathway indicating autophagy initiation. However, chloroquine, an autophagy inhibitor, significantly potentiated the cytotoxic effect of ZK-CH-11d in MDA-MB-231 cells indicating that autophagy is not principally involved in the antiproliferative effect of ZK-CH-11d. Taking together the results from our experiments, we assume that autophagy was activated as a defense mechanism in treated cells trying to escape from chalcone-induced harmful effects.

## 1. Introduction

In 2020, the incidence of breast cancer (BC) worldwide exceeded the incidence of lung cancer and became the most diagnosed type of cancer (11.7% of all newly diagnosed cases) in both sexes and ranks sixth among death-causing cancers [1]. On the other hand, introduction of new, effective chemotherapy significantly increased the 5-year survival rate of BC patients [2]. However, the use of these drugs is often associated with unacceptable toxicity [3]. For this reason, several experimental studies have been performed to find more effective and safer anticancer agents to treat BC [4,5,6]. Nature has played an important role in the treatment of various diseases. Active substances isolated from plants have been shown to exhibit different biological effects including anti-inflammatory, anti-malarial, antioxidant, antituberculotic, antibacterial, antifungal, antiviral, neuroprotective, antidiabetic, and many others [7,8,9]. Additionally, several lines of evidence indicated that various phytochemicals also possess antiproliferative and anticancer activity [10,11,12]. Chalcones, precursors in the synthesis of flavonoids and isoflavonoids in plants, have attracted attention due to their biological activity. Among others, the anticancer effect of both natural and synthetic chalcones has been intensively studied in the last decade [13,14,15,16]. Although the mechanism of the antiproliferative action of chalcones has not yet been fully elucidated, numerous studies documented pleiotropic mode of action [17,18,19,20] including cell cycle arrest, DNA damage, induction of reactive oxygen species production, modulation of various signaling pathways, etc. [19,20,21,22,23,24,25,26]. Furthermore, induction of both intrinsic and extrinsic apoptosis pathways associated with caspases activation, modulation of the Bcl-2 family protein activity, impairment of mitochondrial function with subsequent release of proapoptotic factors such as cytochrome *c* has also been experimentally proven [27,28]. Some of the above-mentioned effects have also been demonstrated by experiments from our laboratory [14,29,30]. Another mechanism of chalcone-induced cell death is autophagy, which may be an alternative way to treat apoptosis-resistant cancers [29]. Autophagy is triggered in response to DNA damage, oxidative stress, nutrient, and growth factor deficiency. Via “recycling” of cytoplasmic contents and cellular homeostasis control, autophagy plays an important role in cytoprotection and is generally recognized as a cell pro-survival mechanism. However, autophagy, under some circumstances, may lead to activation of the apoptosis pathway—because of excessive degradation of basic cell components [30]. It has been revealed that there is an interplay between autophagy and apoptosis. Under different conditions, autophagy can both either antagonize or induce apoptosis [31].

This study investigated the mechanism of the indole chalcone ZK-CH-11d on the induction of apoptosis and autophagy in BC MDA-MB-231 and MCF-7 cells. Our results indicate that the antiproliferative effect of chalcone ZK-CH-11d is associated with induction of apoptosis, cell cycle arrest in the G2/M phase, DNA damage, and inhibition of the PI3K/Akt/mTOR pathway and autophagy.

## 2. Materials and Methods

### 2.1. Tested Compound

(2*E*)-1-(2-fluorophenyl)-3-(2-propoxy-1*H*-indol-3-yl)prop-2-en-1-one indole chalcone (ZK-CH-11d, Figure 1) was synthesized by Zuzana Kudlickova. The structure of the compound was confirmed by using ^1^H, ^13^C nuclear magnetic resonance (NMR), infrared (IR) spectroscopy, and mass spectrometry. The results of these analyzes are presented in Appendix A. The studied indole chalcone was dissolved in DMSO. The final concentration of DMSO in the culture medium was less than 0.2% and showed no cytotoxicity.

### 2.2. Cell Culture

MCF-7 (human breast adenocarcinoma), MDA-MB-231 (human triple-negative breast carcinoma), and MCF-10A (human mammary epithelial cells) cells were used for this study and purchased from ATCC (American Type Culture Collection, Manassas, VA, USA). MCF-7 cells were cultured in DMEM (Dulbecco’s Modified Eagle’s Medium) growth medium (GE Healthcare, Piscataway, NJ, USA) and MDA-MB-231 cells were cultured in RPMI 1640 growth medium (Biosera, Kansas City, MO, USA). Both growth media were supplemented with 10% fetal bovine serum (FBS) and antibiotic/antifungal solution 1 × HyClone^TM^ (GE Healthcare, Chicago, IL, USA). The MCF-10A cell line was cultured in DMEM F12 high glucose medium (Dulbecco’s Modified Eagle’s Medium F12) (Biosera, Kansas City, MO, USA). This medium was supplemented with 10% fetal bovine serum, antibiotic/antifungal solution 1 × HyClone^TM^ (GE Healthcare, Chicago, IL, USA), EGF (final 20 ng/mL), hydrocortisone (0.5 μg/mL final), and insulin (10 μg/mL final). The cells were cultured at 37 °C in an atmosphere containing 5% CO_2_ in humidified air. Prior to each experiment, we performed a cell viability assay using trypan blue dye and the cell viability was greater than ≥95%.

### 2.3. MTT Assay

To determine the half-maximal inhibitory concentration (IC_50_) and to verify the antiproliferative effect of ZK-CH-11d, the MTT colorimetric assay was used. Cells (MDA-MB-231, MCF-7, and MCF-10A) were seeded in 96-well microplates (SARSTEDT, Nümbrecht, Germany) at a density of 5 × 10^3^ cells per well. Twenty-four hours after cells seeding, the chalcone was added at concentrations of 100, 50, and 10 µmol/L and chloroquine at concentrations of 5, 10, 20, 50, and 100 µmol/L. After 24, 48, and 72 h of chalcone and chloroquine treatment, the cells in each well were incubated for 4 h with 10 μL of MTT (5 mg/mL, Sigma-Aldrich Chemie, Steinheim, Germany) at 37 °C in the dark. During this incubation, MTT was metabolized to insoluble formazan by mitochondrial oxidoreductases in cells. 100 μL of SDS (10% sodium decyl sulfate) was added to each well to dissolve the formazan crystals, and the cells were incubated for another 12 h. An automated Cytation™ 3 Cell Imaging Multi-Mode Reader (Biotek, Winooski, VT, USA) was used to measure absorbance at 540 nm. Three independent experiments were performed for each test.

### 2.4. BrdU Cell Proliferation Assay

MDA-MB-231 and MCF-7 cells were seeded in 96-well plates at a density of 5 × 10^3^ and incubated for 24 h in growth medium. After 24 h, ZK-CH-11d was added to the cells at concentrations ranging from 25–55 μmol/mL. After 48 h, BrdU labeling solution was added to the cells and the cells were allowed to incubate. After 24 h, a cell proliferation ELISA, BrdU (colorimetric) kit (Roche Diagnostics GmbH, Mannheim, Germany) was used; the cells were fixed with fixative solution and incubated with anti-BrdU peroxidase conjugate solution for 90 min in the dark at room temperature. The cells were then washed with wash solution (3×) and incubated for 5 min with TMB substrate solution. The color of the solution in the wells of the plate changed to blue. A solution of H_2_SO_4_ (25 μL, 1M) was used to stop the ongoing reaction; the color of the solution changed to yellow. The change in absorbance was analyzed at 450 nm (for yellow, 690 nm for blue) using the automated Cytation^TM^ 3 Cell Imaging Multi-Mode Reader (Biotek, Winooski, VT, USA). Three independent experiments were performed.

### 2.5. Flow Cytometric Analysis

#### 2.5.1. Analysis of Apoptosis

After seeding the cells (MDA-MB-231 and MCF-7) at a density of 1 × 10^5^, the cells were incubated with the tested chalcone ZK-CH-11d at a final concentration of 40 μmol/L, and DMSO at the same concentration was used as a negative control. Subsequently, cells were harvested after 24, 48, and 72 h of treatment, washed in PBS, and incubated in the dark for 15 min in staining solution containing Annexin V-Alexa Fluor^®^ 647 (1:300, Thermo Scientific, Rockford, IL, USA). Subsequently, the cells were washed and incubated with PI (propidium iodide, Sigma-Aldrich, Steinheim, Germany) at a final concentration of 25 μg/mL. A BD FACSCalibur flow cytometer (Becton Dickinson, San Jose, CA, USA) was used for analysis. The analysis was performed in triplicate.

#### 2.5.2. Cell Cycle Analysis

MCF-7 and MDA-MB-231 cells were seeded at a density of 1 × 10^5^/dish in a Petri dish and incubated in growth medium. After 24 h of incubation, test compound ZK-CH-11d was added at a concentration of 40 μmol/L and DMSO at the same concentration as the negative control for all three incubation times. After 24, 48, and 72 h, MCF-7 and MDA-MB-231 cells were harvested, washed with cold PBS, fixed in cold 70% ethanol, and stored at −20 °C for at least 24 h. Prior to analysis, cells were washed again with PBS and resuspended in staining solution in which they were incubated for 30 min in the dark at room temperature. The staining solution contained Triton X-100 (final concentration 0.1%), ribonuclease A (final concentration 0.5 mg/mL) and PI (final concentration 0.025 mg/mL, Sigma-Aldrich). The analysis was performed with a FACSCalibur flow cytometer (Becton Dickinson). The analysis was performed in triplicate.

#### 2.5.3. Flow Cytometric Analysis of Apoptosis and Autophagy Related Proteins

Cells were seeded 1 × 10^6^ on a Petri dish. Twenty-four hours after seeding and cultivation, ZK-CH-11d was added at a final concentration of 40 μmol/L, and DMSO at the same concentration for each exposure time was used as a negative control. MCF-7 and MDA-MB-231 cells were harvested after 24, 48, and 72 h of treatment, washed with PBS, and divided for each assay. Prior to analysis of each protein, cells were incubated for 15 min in staining solution in the dark at room temperature (RT). The antibodies used and their dilutions are listed in Table 1. Analyzes were performed using a FACSCalibur flow cytometer (Becton Dickinson, San Jose, CA, USA).

### 2.6. Western Blot

The cells (MDA-MB-231, MCF-7, and MCF-10A) were incubated with ZK-CH-11d 24, 48, and 72 h, scraped on an ice plate, washed with cold PBS, centrifuged, and isolated with cold LaemLi lysis buffer. The lysis solution contained 1 M Tris/HCl (pH 6.8), glycerol, 20% SDS (sodium dodecyl sulfate), deionized H_2_O, and protease and phosphatase inhibitors. A colorimetric Pierce^®^ BCA protein assay kit (Thermo Scientific, Rockford, IL, USA) was used to evaluate protein concentration, and absorbance was measured at 570 nm using an automated Cytation™ 3 Cell Imagination multimode sensor (Biotek). Protein separation (25–40 µg of sample per well) was performed on an SDS-PAA gel (12%) at 100 V for 3 h and proteins were transferred to polyvinylidene difluoride (PVDF) membranes by iBlot dry blotting (Thermo Scientific). Membranes were blocked in a solution of 5% BSA (bovine serum albumin, SERVA, Heidelberg, Germany) or non-fat dry milk (Cell Signaling Technology^®^) in TBS-Tween (pH 7.4) for 1 h at room temperature. After blocking non-specific binding, the membranes were incubated with primary antibodies (Table 2) overnight at 4 °C. After 24 h, the membranes were washed in TBS-Tween (3 × 5 min) and incubated with horseradish peroxidase-conjugated secondary antibody for 1 h at room temperature. After 1 h, the membranes were washed again in TBS-Tween (3 × 5 min). ECL chemiluminescent substrate (Thermo Scientific, Rockford, IL, USA) and MF-ChemiBIS 2.0 Imaging System (DNR Bio-Imaging Systems, Jerusalem, Israel) were used to detect protein expression. Densitometric analysis was performed using Image Studio Lite software (LI-COR Biosciences, Lincoln, NE, USA) and beta-actin was used as a gel loading control. Results from densitometric analysis are presented in Appendix A. Analyzes were performed as three independent experiments.

### 2.7. Fluorescence Microscopy

The MDA-MB-231 and MCF-7 cells were seeded on 22 mm^2^ coverslips in 6-well plates (SARSTEDT) at a density of 5 × 10^4^/cm^2^ and allowed to grow until the next day. After 24 h, chalcone ZK-CH-11d was added at a concentration of 40 μmol/L. After 24, 48, and 72 h incubation with chalcone and DMSO as a negative control, cells were washed with PBS and fixed with 2% paraformaldehyde (pH 7.2) for 10 min, washed with PBS, and permeabilized with 0.1% Triton X-100 solution for 10 min and washed again with PBS at room temperature. Cells on slides were blocked with Swine serum at a 1:30 dilution. After blocking non-specific binding, cells were incubated with LC3A/B primary antibody (Cell Signaling Technology^®^) for 90 min. Coverslips were washed (3 × 5 min) and incubated with Alexa Flour 594 Goat anti-Rabbit IgG (H + L) secondary antibody (Thermo Scientific, Rockford, IL, USA) diluted according to the manufacturer’s instructions. Cell nuclei were stained with DAPI (Sigma-Aldrich, Steinheim, Germany) at a final concentration of 30 ng/mL in PBS. Coverslips were mounted using Vectashield mounting medium (Cole-Parmer, Vernon Hills, IL, USA). The slides were analyzed using a Nikon Eclipse 90i fluorescence microscope (Nikon, Tokyo, Japan).

### 2.8. Detection of Apoptosis Using an AO/PI Staining

The MDA-MB-231 and MCF-7 cells were seeded in 6-well plates (SARSTEDT) at a density of 5 × 10^4^/cm^2^. After 24 h incubation with growth medium, chalcone ZK-CH-11d was added at a concentration of 40 μmol/L and DMSO at the same concentration. After 24, 48, and 72 h of treatment, the cells were washed in PBS and fixed with 4% paraformaldehyde (pH 7.2) for 30 min. After paraformaldehyde removal, cells were washed with PBS and incubated with a staining solution containing acridine orange (final 10 µg/mL) and propidium iodide (final 10 µg/mL, Sigma-Aldrich) for 1 h at room temperature in the dark. After incubation, the plates were washed again with PBS, dried, and analyzed using an automated Cytation™ 3 Cell Imaging Multi-Mode Reader (Biotek, Winooski, VT, USA).

### 2.9. Statistical Analysis

Results were expressed as the mean ± standard deviation (SD). Statistical analyses of the data were performed using standard procedures with one-way analysis of variance (ANOVA) followed by a Bonferroni multiple comparison test. Differences were considered significant when *p* < 0.05. Throughout this document * determines *p* < 0.05, ** *p* < 0.01, *** *p* < 0.001 vs. vehicle (DMSO).

## 3. Results

### 3.1. Effect of ZK-CH-11d on Cell Proliferation

The effect of synthetic indole chalcone ZK-CH-11d on the proliferation of MDA-MB-231, MCF-7, and MCF-10A cells was evaluated by the MTT assay. For cancer cell lines, the IC50 was 34.03 ± 3.04 and 37.32 ± 1.51 μmol/L (Table 3) after 72 h. The number of viable cells (MDA-MB-231 and MCF-7) after chalcone treatment was significantly reduced after 24 h with a continuing trend after 48 and 72 h for both cell lines. DMSO was used as a negative control and it had no effect on cell growth after 24, 48, and 72 h (Figure 2A,B). For further experiments, we decided to use a concentration of 40 μmol/L in both tumor lines and DMSO was used at a non-toxic concentration of 0.08%.

Results are reported as the mean ± SD of three independent experiments.

### 3.2. BrdU Assay

The incorporation of BrdU (5-bromo-2′-deoxyuridine) into DNA as a thymidine analog during cell replication is used as a sensitive method for detecting DNA synthesis in the S phase of the cell cycle. The BrdU assay showed that chalcone suppressed cancer cells proliferation concentrations ranging from 25 μmol/L to 55 μmol/L, compared to the untreated control. After 72 h of incubation with the tested chalcone, the IC_50_ was 31.66 ± 0.18 μmol/L (MDA-MB-231) and 32.17 ± 0.11 μmol/L (MCF-7) (Figure 3A,B).

### 3.3. Effect on Cell Cycle and Cell Cycle Related Proteins

#### 3.3.1. Cell Cycle Analysis

To determine the mechanisms by which chalcone ZK-CH-11d inhibits cell proliferation, we examined the cell cycle progression in MDA-MB-231 and MCF-7 cells by flow cytometry. In the MDA-MB-231 and MCF-7 cell lines, the tested chalcone caused an increase in the number of cells in the G2/M phase at all three exposition times (24, 48, and 72 h) (Figure 4A,B). We also observed a reduction in the number of cells in the G1 phase in both cell lines. On the other hand, a slight but significant increase in the S phase of the cell cycle was observed in the MCF-7 cell line after 48 and 72 h of incubation. Furthermore, there was also a significant increase in cells with subG0/G1 DNA content (the marker of apoptosis), in both cancer cell lines (Figure 4A,B). These results suggest that the antiproliferative effect of chalcone ZK-CH-11d may be associated with G2/M arrest and apoptosis induction in human breast cancer MDA-MB-231 and MCF-7 cells.

#### 3.3.2. Effect on Cip/Kip Proteins and Cell Cycle Regulating Proteins

p53 is an important regulatory protein with tumor suppressor activity. Activated p53 plays a key role in the response to stress stimuli such as oncogenic activation and DNA damage. DNA damage induces its phosphorylation, increases stability and activity, by inhibiting the interaction with its negative regulator. Chalcone ZK-CH-11d caused a significant increase in the expression of both total and phosphorylated forms of p53 in MDA-MB-231 and MCF-7 cells at all times of treatment (Figure 5A,B).

The p21^Waf1/Cip1^ protein is a tumor suppressor with an important role in cell cycle regulation, the mechanism of action of which is the inhibition of all cyclin–CdK complexes. Cell cycle blockade in the G1/S and G2/M phases results from increased p21 expression following p53 phosphorylation in response to DNA damage. By Western blot analysis, we observed a significant increase in p21 expression in MDA-MB-231 cells after ZK-CH-11d treatment at all times of treatment. Paradoxically, in MCF-7 cells, the studied chalcone was found to cause a decrease in p21 expression at 24, 48, and 72 h after chalcone treatment (Figure 6A,B and Appendix A).

The p27^Kip1^ protein, together with p21, belongs to a family of cyclin-dependent kinase inhibitors whose role is to arrest the cell cycle. Its functions include the regulation of cell proliferation, cell differentiation, and apoptosis. Figure 6A shows that chalcone ZK-CH-11d had a significant effect on upregulation of p27 in MDA-MB-231 cells, but in MCF-7 cells caused a decrease in the expression of this protein in a time-dependent manner (24, 48, and 72 h) (Figure 6A,B and Appendix A).

Wee1 protein kinase is a regulatory factor responsible for the phosphorylation and thus inactivation of cdc2 kinase, which ensures the transition of cells from the G2 phase of the cell cycle to mitosis. Phosphorylation of Wee1 at Ser642, provided by Akt kinase, leads to its translocation to the cytoplasm and a decrease in its activity. By Western blot protein analysis, we observed a significant reduction in the expression of the phosphorylated form of Wee1, in MDA-MB-231 cells at all three times of chalcone treatment (24, 48, and 72 h) and in MCF-7 cells phospho-Wee1 downregulation after 48 and 72 h (Figure 6A,B and Appendix A).

Chk1 kinase acts as a protein with a tumor suppressor effect and plays an important role in controlling the checkpoint in response to DNA damage. Chk1, activated by phosphorylation by ATM/ATR kinases, blocks cdc2 activation, mitotic spindle formation, chromatin condensation, and cell transition into mitosis. As shown in Figure 6A,B and Appendix A, the tested compound significantly induced Chk1 phosphorylation in MDA-MB-231 cells after 24, 48, and 72 h and a significant decrease in MCF-7 cells after all three exposition times.

#### 3.3.3. Inhibition of Cyclins, cdc2 and Rb

Cyclins, together with cyclin-dependent kinases, are responsible for the passage of cells through cell cycle phases and their division. Cyclin B1 forms a complex with CdK1 being involved in the G2/M phase transition of the cell cycle. Phosphorylation of cyclin B1 induces its transition and accumulation in the nucleus and the initiation of mitosis. In some types of cancer, this cyclin may be overexpressed and is therefore associated with a poor prognosis. In our experiment, chalcone ZK-CH-11d caused significant downregulation of phosphorylated cyclin B1 in MDA-MB-231 cells after 24 h of treatment with a maximum after 72 and in MCF-7 cells after 48 and 72 h of treatment (Figure 6C,D and Appendix A).

The target of the activated cyclin D1-CdK1 complex is the phosphorylation of the Rb protein, which releases a transcription factor regulating the expression of proteins re-quired for G1/S phase cell transition. As our experiment showed, the studied compound induced a significant downregulation of cyclin D1 and its phosphorylated form in both tumor cell lines (MBA-MB-231 and MCF-7) during the whole exposure period (from 24 to 72 h) (Figure 6A,B and Appendix A).

Phosphorylation of cdc2 on Tyr15 results in a decrease in its activity, which prevents the progression of the cell cycle process and the entry of cells into mitosis. Surprisingly, exposure of both MBA-MB-231 and MCF-7 cells to chalcone ZK-CH-11d resulted in a considerable decrease in levels of phosphorylated cdc2 (Figure 6C,D and Appendix A).

Rb is a tumor suppressor protein whose function is to regulate cell proliferation and its passage through a restriction point in the G1 phase of the cell cycle. Phosphorylation of the Rb protein results in the release of the E2F transcription factor, which affects the transcription of other proteins required for cell cycle progression. As shown in Figure 6A,B and Appendix A, chalcone ZK-CH-11d significantly inhibited Rb phosphorylation compared to the control, thus inducing an increase in its inhibitory activity at all exposure times in both tumor cell lines (MDA-MB-231 and MCF-7).

Tubulins are key proteins for the smooth transition of cells through the cell cycle, mitotic spindle formation, and mitosis. The key proteins are alpha and beta-tubulin, which interact to form microtubules essential for the cytoskeleton, cell transport, cell division, and other processes. Inhibition of this interaction can lead to a cell cycle block in the G2/M phase. Chalcone ZK-CH-11d altered the expression of α, α1c tubulin, and β tubulin in MDA-MB-231 cells. In particular, there was a significant downregulation of β tubulin after treatment in a time-dependent manner (Figure 6C and Appendix A). Similarly, MCF-7 cells showed a significant reduction in β tubulin expression after 48 and 72 h of chalcone treatment (Figure 6D and Appendix A).

### 3.4. ZK-CH-11d Induces Apoptosis

#### 3.4.1. Apoptosis Analysis with Fluorescence Microscopy

Acridine orange (AO) and propidium iodide (PI) are nucleic acid-binding dyes. Due to the double staining with these dyes, it is possible to divide the cells into populations of living cells, apoptotic or necrotic cells. AO binds to living and dead cells and PI to cells with damaged membranes. As shown in Figure 7A,B, the tested chalcone ZK-CH-11d caused a significant reduction in the number of MDA-MB-231 and MCF-7 breast tumor cell lines. Furthermore, already after 24 h of treatment with a time-dependent trend, chalcone induced a significant increase in the incidence of yellow and orange cells, which together with other results (Annexin V/PI staining, an increase in the cell population in the subG0/G1 phase) suggests that the test compound induces apoptotic cell death of cancer cells.

#### 3.4.2. Annexin V/PI

In cells with an intact membrane, phosphatidylserine (PS) is localized on the inner leaf of the lipid bilayer. Its externalization is considered to be a marker of apoptosis at an early stage. Annexin V and PI double staining is used to distinguish between living cells (An−/PI−), cells in early apoptosis (An+/PI−), late stage of apoptosis/necrosis (An+/PI+), and dead cells (An−/PI+). Annexin V is able to bind PS on the cell surface and PI diffuses through membranes with lost integrity. We observed significant changes induced by the chalcone ZK-CH-11d in both breast tumor lines. MDA-MB-231 cells showed a significant increase in cells in the early stage of apoptosis (after 24 h of treatment) as well as in the late stage of apoptosis (after 48 and 72 h of treatment) (Figure 8A). Similarly, in MCF-7 cells, the number of An+/PI− stained cells increased after 24, 48, and 72 h of treatment and after 24 and 48 h of Annexin V and PI-positive cells (Figure 8B). Dead cells were also found to be significantly increased in both cell lines.

#### 3.4.3. Effect of Chalcone on Mitochondrial Functions

##### Effect on Bcl-2 Family Proteins

The Bcl-2 family of proteins is involved in cell survival and cell death. The antiapoptotic protein Bcl-2, located on the outside of mitochondria, prevents the release of cytochrome *c* and other proapoptotic factors from mitochondria. In many studies, changes in Bcl-2 family protein levels are accompanied by induction of apoptosis. Using Western blot analysis, we observed an increase in the amount of proapoptotic protein Bax in MDA-MB-231 after 72 h of ZK-CH-11d treatment and its paradoxical decrease in MCF-7 cells. The level of Bcl-xL antiapoptotic protein increased in MDA-MB-231 cells after 72 h of treatment (Figure 9A,B and Appendix A).

##### Effect of Chalcone on Expression of Survivin

Survivin belongs to a family of inhibitors of apoptosis, which is significantly ex-pressed in tumor tissue. Its function is to inhibit the caspase cascade, control the G2/M checkpoint of the cell cycle, and inhibit caspase-dependent cell death. Inhibition of phosphorylation of this antiapoptotic protein leads to loss of its function and aids in the propagation of the apoptotic stimulus. Our results showed a significant reduction in phosphorylated survivin in MDA-MB-231 in all three exposition times and in MCF-7 cells after 72 h of treatment (Figure 9A,B and Appendix A).

##### Cytochrome c

As a result of a decrease in MMP and outer mitochondrial membrane damage, cytochrome *c* and other apoptotic factors are released from the intermembrane space into the cytosol. Due to the action of the tested chalcone, there was a massive release of cytochrome *c* in both MDA-MB-231 and MCF-7 cells at 24, 48, and 72 h compared to DMSO as a negative control (Figure 10A,B). These results suggest the involvement of mitochondrial damage and the release of cytochrome *c* on ZK-CH-11d-induced apoptotic cell death.

##### Mitochondrial Membrane Potential (MMP)

Mitochondria are key cell organelles involved in the induction of apoptotic cell death. The difference in electrical potential between the outer and inner mitochondrial membrane, called the mitochondrial membrane potential (ΔΨm), is a key indicator of mitochondrial activity. The cationic dye TMRE, which specifically binds to active, negatively charged mitochondria, is widely used to identify cells that have lost mitochondrial membrane potential. The results of our experiment show that chalcone ZK-CH-11d caused a significant increase in the number of MDA-MB-231 and MCF-7 cells with reduced MMP at all three times (24, 48, and 72 h) of chalcone treatment (Figure 10C,D).

##### Chalcone Induces Activation of Caspases and PARP Cleavage

Mitochondrial damage and the release of proapoptotic factors into the cytosol trigger the formation of apoptosome and lead to the activation of caspases. Caspases are cysteine proteases involved in apoptosis in its initiation and execution phases. In our experiments, treatment with chalcone ZK-CH-11d led to the activation of caspases 3 and 7 (Figure 11). In MDA-MB-231 cells, chalcone induced the cleavage of pro-caspase 3 to the active form of the enzyme in a time-dependent manner after 24, 48, and 72 h of exposure (Figure 9A, Figure 11A and Appendix A). Because MCF-7 cells do not express caspase 3, we evaluated changes in caspase 7 levels in this line. Similar to flow cytometric analysis, Western blot analysis showed a significant increase in activated caspase 7 levels in MCF-7 cells with a maximum after 72 h of chalcone treatment (Figure 9A, Figure 11A, and Appendix A). Activation of these two caspases leads to proteolytic cleavage of the repair enzyme poly-(ADP-ribose)-polymerase (PARP). Inactivation of PARP prevents the repair of damaged DNA, so we further monitored the effect of the tested chalcone on this enzyme. As shown in Figure 9A and Figure 11A,C,D, a time-dependent increase in the levels of the PARP cleavage form was observed in both cell lines. Western blot analysis of the proteins also showed that ZK-CH-11d-induced reduction in the active form of PARP certainly occurred after 24, 48, and 72 h (Figure 11C,D and Appendix A).

### 3.5. ZK-CH-11d Induces DNA Damage

#### 3.5.1. 8-oxo-7,8-Dihydroguanine

8-oxo-7,8-dihydroguanine, or 8-oxoguanine, is an oxidation product of guanine and is widely used as a marker of oxidative stress in vivo and in vitro. In our study, we monitored 8-oxoguanine levels after treatment with ZK-CH-11d. In MDA-MB-231 and MCF-7 cells, the 8-oxoguanine levels were significantly increased at all exposure times (Figure 12A,B).

#### 3.5.2. ATM

Serine/threonine kinase ATM (ataxia telangiectasia mutated) plays an important role in the regulation of DNA repair, cell cycle checkpoints, and apoptosis. The response to double-stranded DNA breaks is ATM phosphorylation. Proteins that interact with ATM include p53, checkpoint kinases, SMC1, Histone H2A.X, and many more. As shown in Figure 12C,D, chalcone ZK-CH-11d caused a significant increase in the number of cells with the phosphorylated form of ATM protein in MDA-MB-231 and MCF-7 cells after treatment at all exposure times with a maximum after 72 h.

#### 3.5.3. SMC1

The SMC1 (structural maintenance of chromosomes 1) protein belongs to the members of the family of proteins which are key regulators of DNA repair, chromosome condensation, and other processes. SMC1 phosphorylation may be mediated by ATM kinase in response to DNA damage. In our experiment, the studied chalcone was shown to induce SMC1 phosphorylation after 24, 48, and 72 h of treatment as well as at all exposure times (from 24 to 72 h) in MCF-7 cells (Figure 12E,F).

#### 3.5.4. Histone HA2.X

Histone H2A.X is considered a sensitive marker of DNA damage. Its phosphorylation to Ser139 is a rapid cellular response to double-stranded DNA breaks induced by genotoxic stress followed by cell cycle arrest, DNA repair, or cell death. Phosphorylation is mediated by various kinases such as ATM, ATR, and other so-called stress kinases. In connection with the above results, after 48 h with continued up to 72 h the tested chalcone significantly increased the phosphorylation of H2A.X in both MDA-MB-231 and MCF-7 cell lines (Figure 13A,B and Appendix A). All these results suggest the involvement of DNA damage in ZK-CH-11d-induced cell death of breast cancer cells.

### 3.6. Chalcone ZK-CH-11d Modulates Signalling Pathways

The mitogen-activated protein kinase (MAPK) family (Figure 14), which mediates stimuli from the external to the internal environment of the cell in response to growth factors, mitogens, and others, plays an important role in the regulation of cell death. Increased Erk1/2 activation is associated with cell proliferation and increased survival. In our experiment with chalcone, paradoxically, there was a significant increase in the phosphorylated form of Erk1/2 in the MDA-MB-231 line, but we observed a decrease in phospho-Erk1/2 in MCF-7 cells (Figure 14A,B and Appendix A). P38 MAPK is also a stress-activated protein kinase responsible for differentiation, apoptosis, and autophagy. In both tumor cell lines, the chalcone ZK-CH-11d was shown to significantly increase p38 MAPK phosphorylation after 24, 48, and 72 h for MDA-MB-231 and after 72 h for MCF-7 (Figure 14A,B and Appendix A).

### 3.7. ZK-CH-11d Induces Autophagy

#### 3.7.1. Inhibition of PI3K/Akt/mTOR Pathway

Autophagy is regulated by a signaling pathway that upstream involves phosphorylation of PI3K family proteins (Figure 15). PI3Ks are heterodimeric kinases whose phosphorylation is stimulated by hormones and various pro-survival and growth agents involved in processes such as cell proliferation, migration, cell cycle transition, and cell death. PI3K provides phosphorylation of Akt. The major PI3K inhibitor is PTEN. The aim of our experiment was to determine the effect of the chalcone ZK-CH-11d on this signaling pathway. A significant decrease in the phosphorylation of the p85 and p55 subunits of the PI3K kinase at all times was observed in ZK-CH-11d-treated MDA-MB-231 cells (Figure 15A and Appendix A). In the MCF-7 cancer cell line, we observed the same effect in reducing the PI3K phosphorylation. (Figure 15B and Appendix A).

The PI3K downstream interacts with the serine/threonine protein kinase Akt. It is a key regulator of many cellular processes such as cell signaling, cell growth and proliferation, cell cycle, angiogenesis, metabolism, and others. Its phosphorylation at Thr308 by PI3K-dependent kinase propagates an intracellular signal and affects downstream proteins, e.g., mTOR. As shown in Figure 15A and Appendix A, ZK-CH-11d significantly reduced Akt phosphorylation at all treatment times (24 to 72 h) in both tumor cell lines MDA-MB-231 and MCF-7 (Figure 15B and Appendix A).

The mammalian target of rapamycin (mTOR) protein plays an important role in cell metabolism and cell growth and is a major negative control protein in the regulation of autophagy. Since mTOR is a direct substrate for Akt, the results shown in Figure 15A,B and Appendix A are in line with our assumptions. The chalcone studied significantly inhibited mTOR phosphorylation on Ser2448 in both cell lines.

#### 3.7.2. Effect of Chalcone on Expression of Autophagy-Related Proteins

PTEN (phosphatase and tensin homolog deleted on chromosome ten) is the major negative regulator of the PI3K/Akt/mTOR signaling pathway. In many cancers, it is often mutated, and its phosphorylation can lead to loss of its phosphatase activity and tumor suppressor function. The results of our experiment showed that the ZK-CH-11 molecule induced a significant reduction in the levels of phosphorylated PTEN protein in the tumor cell lines MDA-MB-231 and MCF-7 after chalcone treatment from 24 to 72 h. At the same time, we observed a significant increase in the levels of active, unphosphorylated PTEN after 24, 48, and 72 h of treatment (Figure 16A,B and Appendix A).

The beclin-1 protein is important for the recruitment of autophagic proteins into the preautophagosomal structure (PAS), where together with other proteins they form a core complex. Beclin-1 activity and its cleavage by caspases are considered a crosstalk between autophagy and apoptosis. Our experiments show that the tested chalcone induced a significant reduction in beclin-1 protein levels in MCF-7 cells at each time of treatment (24, 48, and 72 h) as well as in MDA-MB-231 cells (Figure 16A,B and Appendix A).

ULK1/Atg1 is a protein essential for the process of autophagy and degradation of cytoplasmic content. Under nutrient-deficient conditions, AMPK is activated, which directly activates ULK1. In contrast, the major negative regulator of ULK1 is the mTOR autophagy inhibitor, which phosphorylates ULK1 on Ser757 and thus inactivates it. As shown in Figure 16A,B and Appendix A, the test compound significantly reduced the levels of phosphorylated ULK1 at this site in MDA-MB-231 cells (after 48 and 72 h of treatment) and MCF-7 for the duration of the chalcone exposure.

AMPK (AMP-activated protein kinase) plays an important role in the response of cells to external stress as an energy sensor and thus participates in the regulation of intracellular energy balance. Phosphorylation of AMPK leads to its activation, which directly interacts with ULK1 and autophagic promotes. The fact that chalcone ZK-CH-11d is involved in the induction of autophagy proves that it caused a significant increase in AMPK alpha phosphorylation in the MDA-MB-231 and MCF-7 lines after treatment from 24 to 72 h (Figure 16A,B and Appendix A).

Atg7 belongs to the family of Atg proteins (autophagy related) involved in the process of autophagy. Atg7 is an E1-like activating enzyme that is essential for the successful conversion of LC3-I (cytosolic form) to its conjugated form LC3-II (conjugated LC3 with phosphatidylethanolamine). As shown in Figure 16A,B and Appendix A, the studied chalcone did not change Atg7 protein expression in the MDA-MB-231 and MCF-7 cell lines.

The p62 protein (Sequestosome 1), considered a marker of autophagy, is recruited to the autophagosomal membrane by interaction with LC3 and is therefore essential for the transition from polyubiquitinated cargo and autophagosome. Lysosomal degradation of autophagosomes leads to a decrease in SQSTM1 levels during autophagy. In our experiments with chalcone ZK-CH-11d, we observed a significant increase in relative p62 expression from 24 to 72 h of exposure in MDA-MB-231 cells and after 48 and 72 h in MCF-7 cells (Figure 17A,B).

The LC3 protein (light chain 3) is an essential part of autophagosome formation, so it is considered a marker of autophagy. By a mechanism involving the Atg3 and Atg7 proteins, LC3A (cytosolic form) is converted to LC3B (conjugated) and associated with autophagic vesicles. Using Western blot analysis of proteins (Figure 16A,B and Appendix A) and fluorescence microscopy (Figure 18A,B), we observed a significant increase in LC3A/B protein expression by chalcone in MDA-MB-231 and MCF-7 cells at all three exposure times. These results suggest that chalcone ZK-CH-11d is significantly involved in the induction of autophagy in breast cancer cells.

### 3.8. Effect of ZK-CH-11d and Chloroquine on Cell Proliferation

A well-known antimalarial drug, chloroquine, is often used to assess the extent to which autophagy is involved in cell death. Chloroquine inhibits autophagy by blocking autophagic flux. This mechanism is explained by the disruption of the fusion of autophagosomes with lysosomes and the influence of the acidity of lysosomes. The data presented in Figure 19A,B show that chloroquine at concentrations from 5 to 100 μmol/L in combination with chalcone ZK-CH-11d significantly decreased MDA-MB-231 cell survival compared to ZK-CH-11d only treatment. This effect has not been found in MCF-7 cells.

## 4. Discussion

Despite the ever-increasing pace of research of new, effective drugs with minimal side effects, nature is the most important source of drugs and patterns for the synthesis of semisynthetic and synthetic drugs. Antitumour effects have been demonstrated in several plants, initiating research into the mechanisms of their antitumour activity [32]. 

As it was mentioned above, chalcones possess broad spectrum of biological activities including anticancer. Moreover, chalcones also provide wide possibilities for substitution and modification of their structure and effect [33,34].

In our experiments, we focused on the study of antiproliferative mechanisms of action of the synthetic, indole chalcone ZK-CH-11d. We compared the activity of the tested molecule on the viability of the breast tumour cell lines MDA-MB-231 and MCF-7 and the non-tumour breast cell line MCF-10A. The MTT metabolic activity assay showed IC50 of 34.03 ± 3.04 μmol/L (MDA-MB-231) and 37.32 ± 1.51 μmol/L (MCF-7). The cytotoxic effect in the non-tumour line MCF-10A at concentrations up to 100 μmol/L was not demonstrated. These results were verified by the BrdU proliferation assay, where the chalcone significantly inhibited the proliferation of both tumour cell lines (Figure 3). 

In our recent review [35] we described pleiotropic mechanisms of antiproliferative effect of chalcones at molecular level. As documented, antiproliferative effect of several chalcones was associated with arrest of cell cycle, mostly at the G2/M phase [14,30,36,37]. In the present study, treatment of human BC cells with chalcone ZK-CH-11d resulted in arrest of cell cycle at G2/M phase, what is often seen after DNA damage. As a consequence of DNA damage, eukaryotic cells start complex signalling pathways resulting in activation of different cellular responses, such as cell cycle arrest or DNA repair [38]. 

Cyclin B1 and cyclin-dependent kinase 1 (also called cdc2) play the key role in G2/M cell cycle regulation [39]. The cdc2/cyclin B complex reacts to DNA damage and prevents cell cycle progression by inhibitory phosphorylation of cdc2 at Tyr-15. This reaction allows DNA repair before cells enter mitosis. In most of experimental works, G2/M arrest in cancer cells has been associated with increased cdc2 phosphorylation [36,37,40]. Surprisingly, our results showed that chalcone ZK-CH-11d significantly decreased cdc2 phosphorylation in both BC cell lines. On the other hand, this phenomenon is not unique because decreased cdc2 phosphorylation and arrest of cell cycle at G2/M phase has also been described previously in different cancer cell lines [41,42,43]. Additionally, we also found significant decrease of phosphorylated form of cyclin B1 in ZK-CH-11d-treated cancer cells. As phosphorylation of cyclin B1 is important for cyclin-B1/cdc2 kinase activation [44], our results indicate that ZK-CH-11d-induced G2/M arrest can be associated with dysfunction of this complex.

Because cyclin B1/cdc2 complex plays a key role in controlling the M phase of cell cy-cle, it is tightly coordinated at multiple levels. A critical function in blocking activation of cyclin B1/cdc2 play p21Waf1/Cip1 and p27Kip1 proteins [45,46]. Western blot analyses showed that both proteins were significantly upregulated in ZK-CH-11d-treated MDA-MB-231 cells. Paradoxically, there was significant reduction in p21Waf1/Cip1 and p27Kip1 in the MCF-7 line. Regarding p21Waf1/Cip1 the same cell-dependent differences have been described recently by Zhang et al. (2020) [47]. In timosaponin AIII-treated cells the G2/M arrest has been associated with upregulation of p21 in MDA-MB-231 cells with its simultaneous downregulation in MCF-7 cells. The author suggests that these differences can by associated with the different status of p53 in these two cell lines. In addition, downregulation of p21Waf1/Cip1 during G2/M arrest has also been reported recently in S adenosyl L methionine-treated oral squamous cancer cell line [48]. These results indicate that mechanism of antiproliferative effect of ZK-CH-11d may be cell type-dependent. 

Another negative regulator of cell cycle is Wee1 kinase that controls the entry of the mitotic phase of the cell cycle by inhibitory effects on the cyclin B1/cdc2 complex [49]. On the other hand, phosphorylation of Wee1 led to its deactivation. In our study we observed that ZK-CH-11d caused significant reduction in the levels of phosphorylated Wee1 on Ser642 in both MDA-MB-231 and MCF-7 cell lines.

In addition to the above-mentioned modulators of cell transition, microtubules, formed from α-tubulin and β-tubulin, play role in different biological processes including cell division [50]. Interaction of drugs with microtubules can disrupt normal cell division due to deficient mitotic spindle formation [51]. Several authors have described the ability of chalcones to interfere with colchicine site and thus inhibit tubulin polymerization and microtubule formation with subsequent cell cycle arrest in mitosis and cell death triggered by apoptosis [52,53]. In our experiment, we found downregulated β-tubulin expression in both tumour lines after treatment in time interval from 24 to 72 h. We suggest that this effect can be associated with inhibition of spindle formation during mitosis followed by G2/M arrest and apoptosis.

In addition to G2/M arrest, cell cycle analysis has also shown significant increase in cell number with subG0/G1 DNA content, a marker of apoptosis, in both cancer cell lines. This observation prompted us to evaluate the ability of chalcone ZK-CH-11d to induce apoptosis.

Mitochondria are subcellular structures with crucial role in the cell life and death. After apoptotic stimuli, the structure and function of mitochondria is seriously changed [54]. Mitochondrial outer membrane permeabilization (MOMP) is connected with decrease of mitochondrial membrane potential (Δψm, MMP) with further release of proapoptotic factors such a cytochrome c, Smac/DIABLO or apoptosis inducing factor to cytosol with subsequent activation of apoptosis machinery [55]. It is well known that proteins of the Bcl-2 family play important role in the MOMP during apoptosis [56]. In the present study we found significant decrease of MMP and upregulated expression of Bax (proapototic member of Bcl-2 protein family) in MDA-MB-231. The level of Bcl-xL (antiapoptotic member of Bcl-2 protein family) has not been changed. An elevated Bax/Bcl-xL ratio favours apoptosis via releasing of proapoptotic proteins from intermembrane space to the cytosol. These proteins include cytochrome c, which activates cytosolic caspase 9. Subsequently, activated caspase 9 can activate effector caspase 3 and caspase 7 [57]. In our study, significant release of cytochrome c has been observed in both ZK-CH-11d-treated BC cell lines. Moreover, activation of caspase 3 in MDA-MB-231 cells and caspase-7 in MCF-7 cells has been detected by western blot analysis. The consequence of this is caspase activation resulting in activation of apoptotic signalling pathway. 

Another marker of apoptosis is the proteolytic cleavage of poly(ADP-ribose)polymerase (PARP), a nuclear enzyme involved in DNA repair. Caspase 3 and caspase 7, cleave the 116-kDa form of PARP to generate a 24-kD and 85-kDa fragments [58]. In the present study, activation of caspase 3 and caspase 7 was associated with PARP cleavage. Similar degradation of PARP by activated caspases has been described in cancer cells treated with natural [59,60] or synthetic chalcone derivatives [14,61]. Finally, apoptosis has also been confirmed by analysis of phosphatidylserine externalization using Annexin V/PI staining. As shown in Figure 8, apoptotic cell death occurred after 24 h of chalcone treatment with a time-dependent trend up to 72 h in both tumour cell lines. 

Furthermore, activation of apoptosis machinery can also be a result of suppression of the members of the inhibitor of apoptosis (IAP) protein family. Survivin, one of the members of this protein family, has an antiapoptotic effect by binding to and inhibiting caspases (3, 7 and 9) and regulating the cell cycle in the G2/M phase [62]. In addition, it has been observed that surviving phosphorylation is required to maintain survivin expression in cancer cells [63]. In our experiment, surviving phosphorylation was significantly decreased in ZK-CH-11d-treated cells. These results are in accordance with several studies in which survivin downregulation was observed in chalcone-induced apoptosis [64,65,66,67]. 

It has been demonstrated that antiproliferative effect of several chalcones was associated with DNA damage [25,68]. In response to DNA damage, the DNA damage response (DDR) system is activated to prevent DNA mutations and maintain genome integrity [69]. During this process, several kinases are activated including ATM (Ataxia telangiectasia mutated). Activation (phosphorylation) of ATM is essential for further phosphorylation of effector proteins such as p53, histone H2A.X, SMC1 and many others [70]. The present study indicates that chalcone ZK-CH-11d-induced G2/M arrest can be a result of ATM pathway activation. Our data showed significant increase of phosphorylated (active) form of ATM, p53, SMC1 and H2AX in both cell lines, implying that studied chalcone blocks cell cycle at G2/M phase via activation of ATM pathway. Furthermore, it has been proved that antiproliferative effect of several chalcones is associated with reactive oxygen species (ROS) generation [71,72,73,74]. Our results showed increased levels of 8-oxo-7,8-dihydroguanine, a marker of oxidative DNA damage, in ZK-CH-11d-treated cells. Although it was not the aim of our study, we suggest that ROS can be involved in DNA damage followed by apoptosis induction.

Our previous studies revealed modulation of several signalling pathways in chalconeinduced apoptosis [14,29,30]. This prompted us to study the effect of ZK-CH-11d on mitogen-activated protein kinase (MAPK) family members, including Erk1/2 and p38 MAPK. The extracellular signal-regulated kinase (Erk1/2) pathway is involved in numerous cell functions including apoptosis or autophagy. This effect is cell type and stimulus-dependent [75]. Activation (phosphorylation) of Erk1/2 pathway generally promotes cell survival and proliferation [76]. However, the proapoptotic function of the active Erk1/2 has been described for apoptosis induced by various compounds including chalcones [77,78]. On the other hand, several chalcones induced apoptosis is associated with downregulated phosphorylation of Erk1/2 [79,80]. We observed a time-dependent increase in Erk1/2 phosphorylation in MDA-MB-231 cells and a decrease in its phosphorylation in MCF-7 cells. Because of controversial role of Erk1/2 in cell proliferation and cell death [81], additional studies are needed to elucidate the function of this kinase in these processes. In addition, we also studied phosphorylation status of p38 MAPK due to fact that activation of this kinase is accompanied with apoptosis induction [82,83,84]. Our experiments have shown that chalcone ZK-CH-11d-induced an increase in p38 MAPK phosphorylation in both cancer lines.

The catabolic process of degradation of macromolecules and cellular organelles known as macroautophagy, hereinafter referred to as autophagy, may also contribute to cell death. Autophagy is generally considered as a survival mechanism. However, under some conditions it has been linked to caspase-independent cell death [85]. The regulation of autophagy in cancer cells is complex. Numerous studies have documented that several signalling pathways such as AMPK or PI3K/Akt/mTOR pathways play a crucial role in autophagy regulation [86,87]. It is well known that mTOR (negative regulator of autophagy) is a downstream molecule of PI3K/Akt pathway and suppression of this pathway triggers autophagy [88]. In addition, inhibition of mTOR activity and activation of autophagy is also associated with activation of AMPK [89]. In the present paper, chalcone ZK-CH-11d suppressed phosphorylation of PI3K, Akt and mTOR with simultaneous up-regulation of AMPK phosphorylation. These results indicate, at least partially, that ZK-CH-11d-induced autophagy in BC cells is mediated via the activation of AMPK with simultaneous inhibition of PI3K/Akt/mTOR pathway. This finding is in line with previous reports [90,91]. Moreover, inhibitory effect of ZK-CH-11d on PI3K/Akt/mTOR pathway can be also associated with PTEN, a negative regulator of PI3K/Akt signalling [92]. There are variety of mechanisms involved in PTEN regulation, of which phosphorylation is the most important. It was recognized that phosphorylation on Ser380, Thr382, and Thr383 residues lead to a loss of phosphatase activity [93]. Our experimental results show that chalcone ZK-CH-11d upregulated expression of PTEN and downregulated its phosphorylated (Ser380) form in both cell lines. 

However, a numerous other proteins are involved in the regulation of autophagy.

In our study we also studied other autophagy-associated proteins such as ULK1 (Unc-51 like autophagy activating kinase 1), LC3A/B (Microtubule-associated protein 1A/1B-light chain 3), p62, Atg7 and beclin-1. ULK1/Atg1 (Autophagy-related protein 1) is a member of an initiation complex involved in the autophagic process. It has been found that phosphorylation of ULK1 Ser 317 and Ser 777 by AMPK directly activates autophagy and, vice versa, mTOR-induced phosphorylation of ULK1 on Ser 757 lead to autophagy suppression [94]. We found that the studied chalcone induced a significant downregulation of ULK1 phosphorylation on Ser757, which can stimulate the autophagy-inducing interaction between ULK1 and AMPK. Furthermore, LC3A/B is also considered to be the key marker of ongoing autophagy. Together with Atg7 it is essential for autophagosome biogenesis [95]. Consistent with the findings of the other authors [77,96] we observed time-dependent upregulation of LC3A/B protein by Western blot analysis and fluorescence microscopy in both breast tumour lines. On the other hand, changes in Atg7 expression in ZK-CH-11d-treated cells have been nonsignificant. Furthermore, our results also showed suppressed expression of beclin-1, a well-known positive regulator of autophagy. However, several experimental data confirmed increased expression of beclin-1 when autophagy was initiated [97,98,99]. Beclin-1 also represents a crosstalk between apoptosis and autophagy through the interactions with survivin, Bcl-2 proteins, and others [100]. Moreover, beclin-1 is a substrate for several caspases including caspase 3 and caspase 7 [101] and beclin-1 cleavage suppresses its proautophagic activity [102]. As we mentioned above, ZK-CH-11d significantly induced activation of both caspase 3 and caspase 7 in treated BC cells and we suggest that decrease of beclin-1 levels can be associated with caspases activation. 

To clarify the role of ZK-CH-11d-induced autophagy on cell death, we analyzed effect of chloroquine, an inhibitor of autophagy [103], on cell viability suppressed by ZK-CH-11d. As MTT assay showed (Figure 19A,B) chloroquine significantly potentiated the cytotoxic effect of ZK-CH-11d in MDA-MB-231 cells. So, we suggest that autophagy is not involved in antiproliferative effect of ZK-CH-11d and it is activated as a defense mechanism in treated cells. Furthermore, as indicated by increased levels of p62, autophagy was not completed probably due to the switch to apoptosis. Similar results have been obtained by Emanule et al. (2018) in litchi fruit extracts-treated human colon cancer cells [104].

In summary, we studied the antiproliferative activity of the indole derivative of chalcones, ZK-CH-11d on BC cells, MDA-MB-231 and MCF-7. Using various methods, we have shown that the chalcone ZK-CH-11d blocked the cell cycle in the G2/M phase and affected proteins associated with cell cycle regulation. We have observed the ZK-CH-11d-induced mitochondrial pathway of apoptosis which was associated DNA damage. The chalcone also triggered an autophagy process and modulated selected signalling pathways.

## 5. Conclusions

Taken together, the results presented here demonstrated that indole chalcone ZK-CH-11d inhibited in vitro proliferation of MDA-MB-231 and MCF-7 BC cells via cell cycle arrest at the G2/M phase and apoptosis induction. Loss of MMP and release of cytochrome *c* indicate the induction of the intrinsic (mitochondrial) apoptosis pathway. Additionally, chalcone ZK-CH-11d treatment led to modulation of the phosphorylation status of several kinases including Erk1/2, p38 MAPK, and Akt. Furthermore, phosphorylation of AMPK and inhibition of the PI3K/Akt/mTOR pathway as well as activation of some autophagy-related proteins indicate that autophagy can also be involved in apoptosis in treated cells. However, results with chloroquine, an autophagy inhibitor, indicate that autophagy is not a component of the antiproliferative effect of ZK-CH-11d and it is activated as a defense mechanism.

Despite the fact that several details of the antiproliferative effects of ZK-CH-11d remain to be elucidated, the results presented here demonstrated that this indole chalcone derivative may be a promising compound for the treatment of BC.

## Figures and Tables

**Figure 1 pharmaceutics-14-00503-f001:**
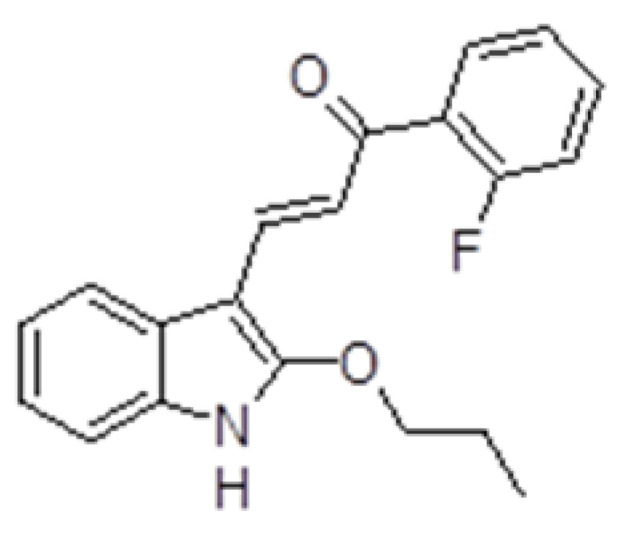
Structure of tested compound ZK-CH-11d.

**Figure 2 pharmaceutics-14-00503-f002:**
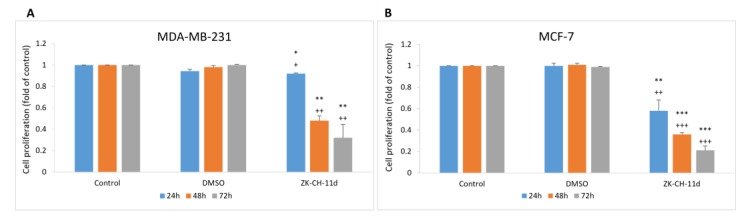
Effect of ZK-CH-11d on the proliferation of MDA-MB-231 (**A**) and MCF-7 (**B**) cells after 24, 48 and 72 h. The data are the results of three independent experiments. Values represent the mean ± standard deviation. Statistical significance: * *p* < 0.05, ** *p* < 0.01, *** *p* < 0.001 vs. DMSO; + *p* < 0.05, ++ *p* < 0.01, +++ *p* < 0.001 vs. untreated cells.

**Figure 3 pharmaceutics-14-00503-f003:**
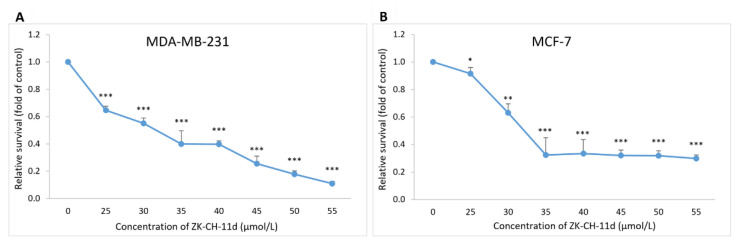
Effect of ZK-CH-11d on BrdU incorporation in MDA-MB-231 and MCF-7 cells. MDA-MB-231 (**A**) and MCF-7 (**B**) cells were exposed to chalcone at concentrations of 25–55 μmol/L for 72 h of treatment. The data show the mean values ± standard deviation of three independent experiments. Statistical significance: * *p* < 0.05, ** *p* < 0.01, *** *p* < 0.001.

**Figure 4 pharmaceutics-14-00503-f004:**
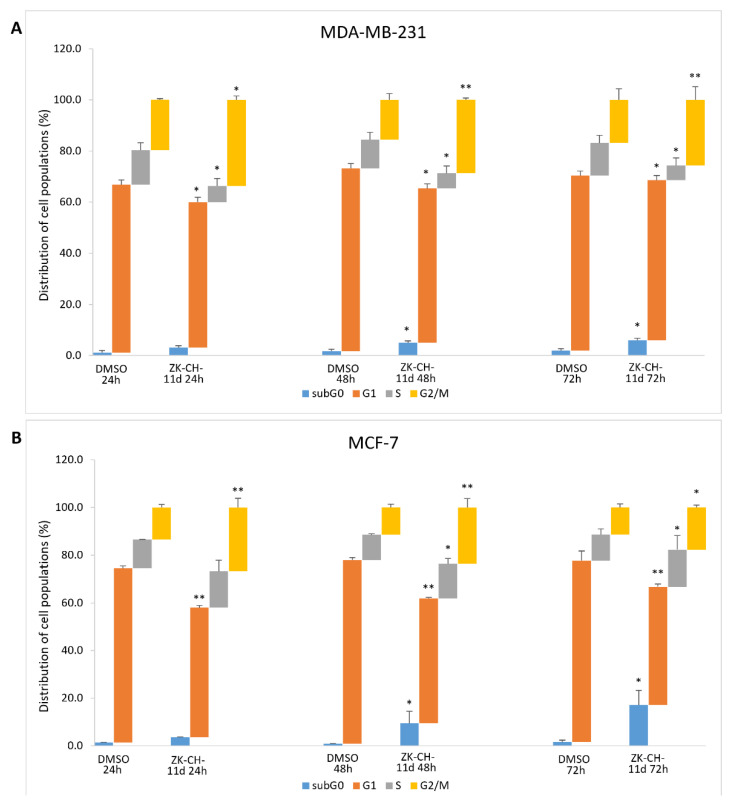
Cell cycle analysis of MDA-MB-231 (**A**) and MCF-7 (**B**) cells after treatment with chalcone ZK-CH-11d 24, 48, and 72 h at a concentration of 40 μmol/L. Values show the mean of three independent experiments. Statistical significance: * *p* < 0.05, ** *p* < 0.01 vs. DMSO (vehicle).

**Figure 5 pharmaceutics-14-00503-f005:**
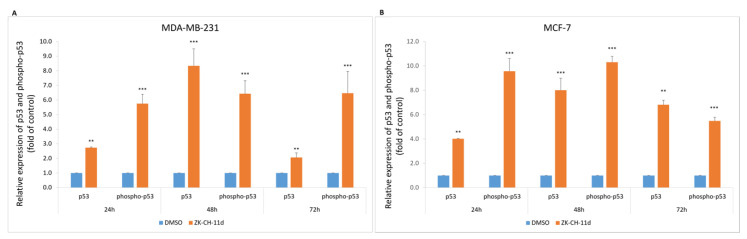
Flow cytometric analysis of p53 protein and its phosphorylated form in MDA-MB-231 (**A**) and MCF-7 (**B**) cells after 24, 48, and 72 h of ZK-CH-11d treatment at a concentration of 40 μmol/L. The values represent the mean of three independent experiments. Statistical significance: ** *p* < 0.01, *** *p* < 0.001 vs. DMSO (vehicle).

**Figure 6 pharmaceutics-14-00503-f006:**
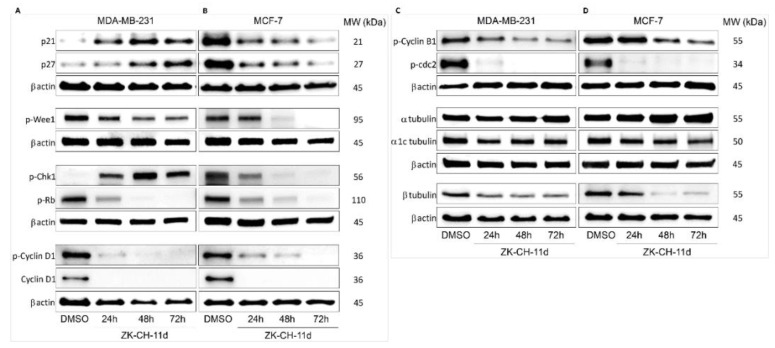
Western blot analysis of cell cycle proteins and related proteins and tubulins in MDA-MB-231 (**A**,**C**) and MCF-7 (**B**,**D**) cells after treatment with chalcone ZK-CH-11d for 24, 48, and 72 h.

**Figure 7 pharmaceutics-14-00503-f007:**
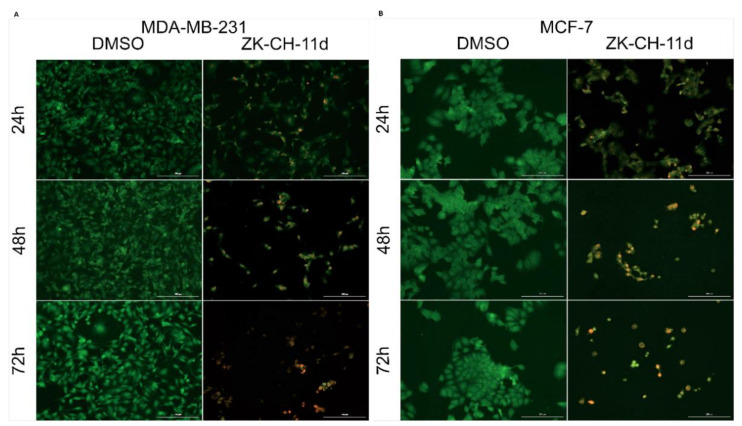
Fluorescence microscopic analysis of apoptosis using Acridine orange/Propidium iodide staining of MDA-MB-231 (**A**) and MCF-7 (**B**) cells after treatment with chalcone ZK-CH-11d at a concentration of 40 μmol/L for 24, 48, and 72 h of treatment. Green represents live cells, yellow cells are in early apoptosis, orange cells are in late apoptosis, and red represents dead/necrotic cells. The figure is a representative average of three independent experiments. Magnification: 100×.

**Figure 8 pharmaceutics-14-00503-f008:**
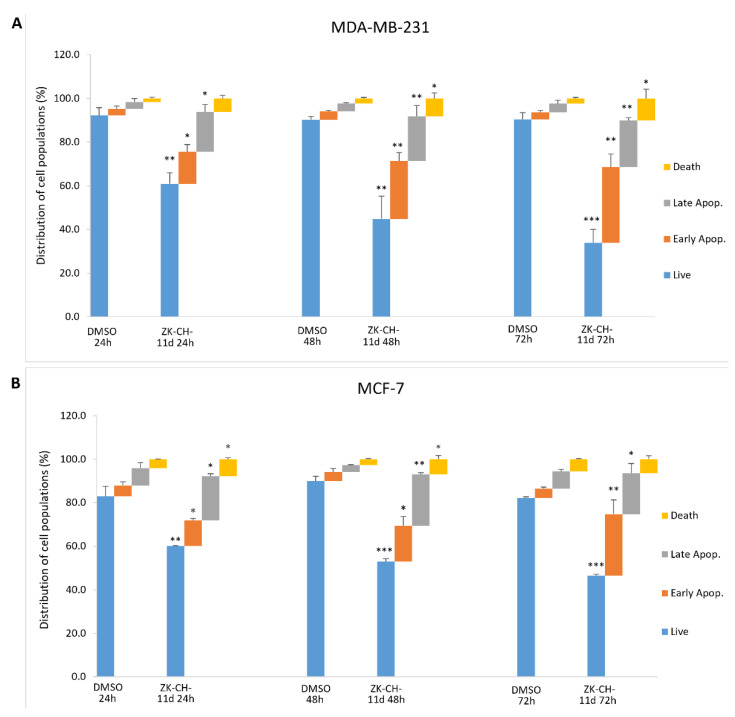
Flow cytometric analysis of ZK-CH-11d-induced apoptosis in MDA-MB-231 (**A**) and MCF-7 (**B**) cells after treatment for 24, 48, and 72 h. Data are presented as the mean of three independent experiments as the mean percentage ± SD. Statistical significance: * *p* < 0.05, ** *p* < 0.01, *** *p* < 0.001 vs. vehicle (DMSO).

**Figure 9 pharmaceutics-14-00503-f009:**
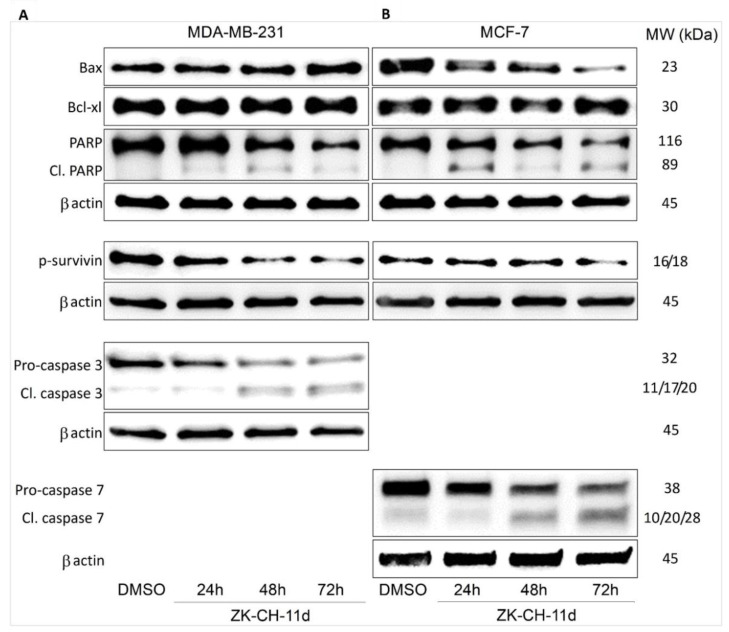
Western blot analysis of apoptosis-related proteins in MDA-MB-231 (**A**) and MCF-7 (**B**) cells after treatment with ZK-CH-11d at a concentration of 40 μmol/L for 24, 48, and 72 h of treatment.

**Figure 10 pharmaceutics-14-00503-f010:**
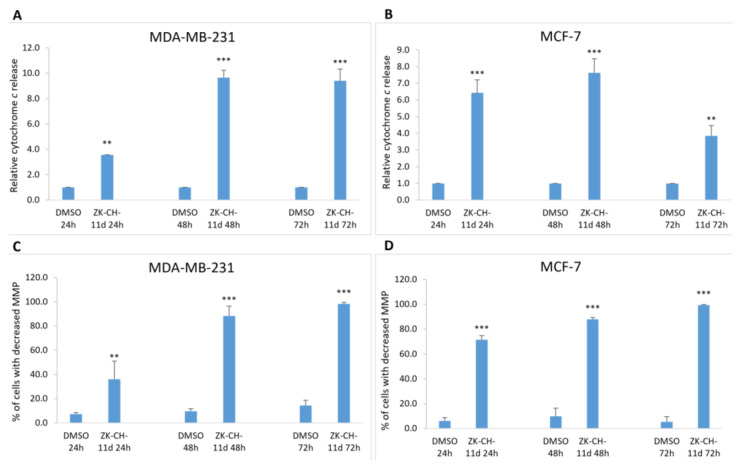
Analysis of changes in mitochondrial functions in MDA-MB-231 and MCF-7 tumor cells after treatment with chalcone ZK-CH-11d at a concentration of 40 μmol/L. Relative cytochrome c release in MDA-MB-231 (**A**) and MCF-7 lines (**B**), and changes in mitochondrial membrane potential (MMP) in MDA-MB-231 (**C**) and MCF-7 (**D**) cells after chalcone treatment after 24, 48, and 72 h. The values shown are the mean of three independent experiments compared to the control (DMSO). Statistical significance: ** *p* < 0.01, *** *p* < 0.001.

**Figure 11 pharmaceutics-14-00503-f011:**
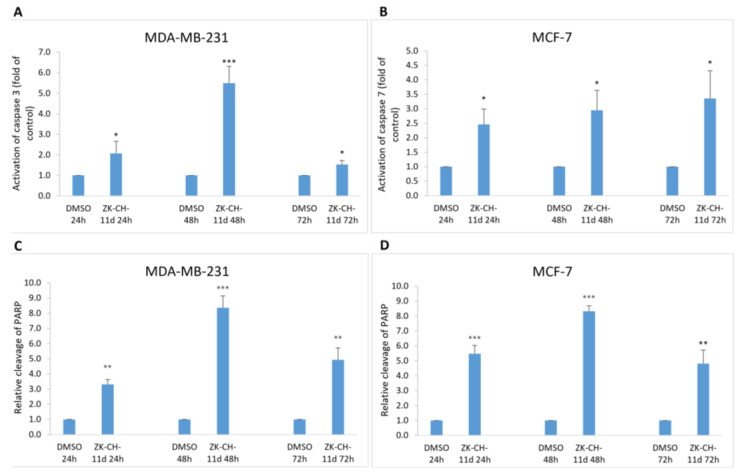
Flow cytometric analysis of caspase activation and induction of PARP cleavage. Relative activation of caspase 3 in MDA-MB-231 cells (**A**) and caspase 7 in MCF-7 (**B**) cells after ZK-CH-11d treatment at 24, 48, and 72 h and relative PARP cleavage in MDA-MB-231 (**C**) and MCF-7 cells (**D**). The values shown are the mean of three independent experiments compared to the control (DMSO). Statistical significance: * *p* < 0.05, ** *p* < 0.01, *** *p* < 0.001.

**Figure 12 pharmaceutics-14-00503-f012:**
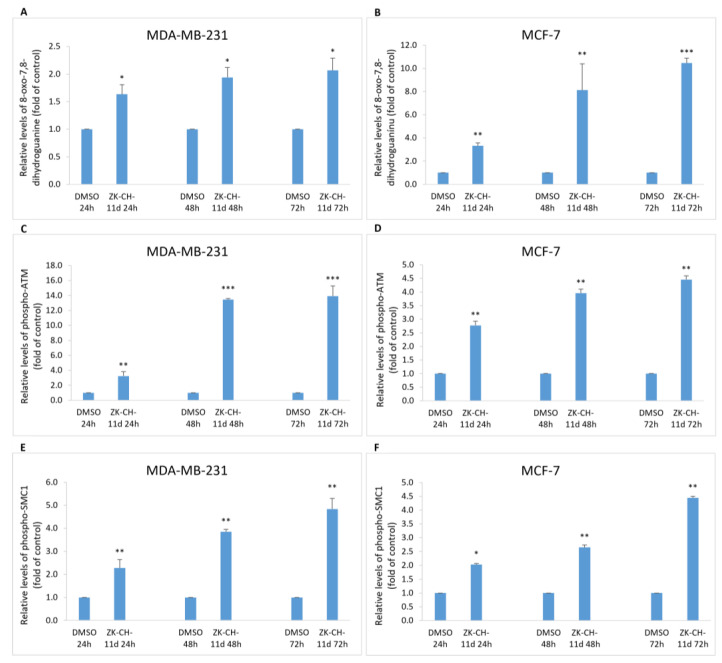
Analysis of DNA damage induced by chalcone ZK-CH-11d at a concentration of 40 μmol/L. Relative changes in 8-oxo-7,8-dihydroguanine levels (**A**,**B**), phospho-ATM (**C**,**D**) and phospho-SMC1 (**E**,**F**) protein levels in MDA-MB-231 and MCF—7 after 24, 48, and 72 h of chalcone treatment. The values are the mean of three independent experiments. Statistical significance: * *p* < 0.05, ** *p* < 0.01, *** *p* < 0.001 vs. vehicle (DMSO).

**Figure 13 pharmaceutics-14-00503-f013:**
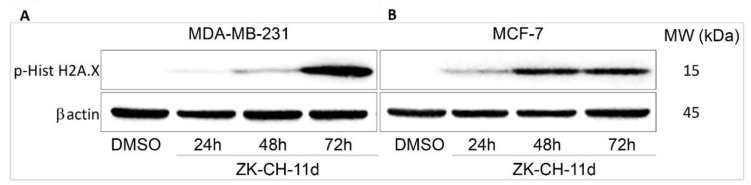
Western blot analysis of H2A.X phosphorylation after 24, 48, and 72 h of ZK-CH-11d treatment in MDA-MB-231 (**A**) and MCF-7 (**B**) cells.

**Figure 14 pharmaceutics-14-00503-f014:**
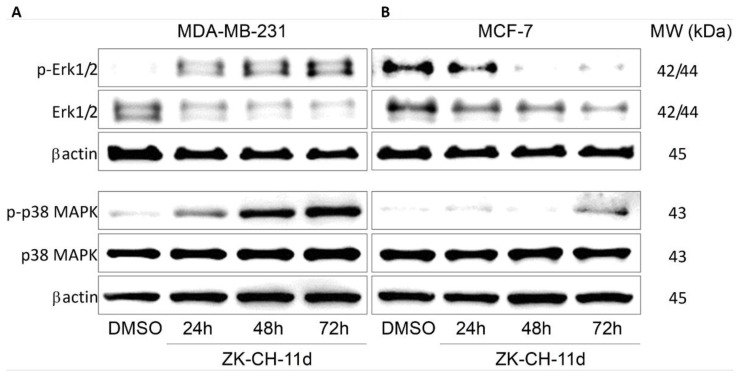
Analysis of Erk1/2 and p38 MAPK signaling pathways after treatment with ZK-CH-11d in MDA-MB-231 (**A**) and MCF-7 (**B**) cells.

**Figure 15 pharmaceutics-14-00503-f015:**
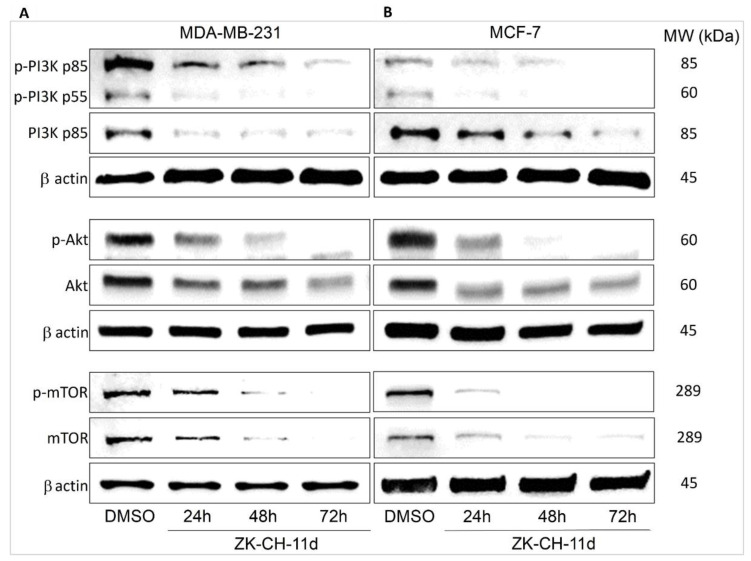
Western blot analysis of PI3K-Akt-mTOR signaling pathway in MDA-MB-231 (**A**) and MCF-7 (**B**) cells after treatment with chalcone ZK-CH-11d for 24, 48, and 72 h of treatment.

**Figure 16 pharmaceutics-14-00503-f016:**
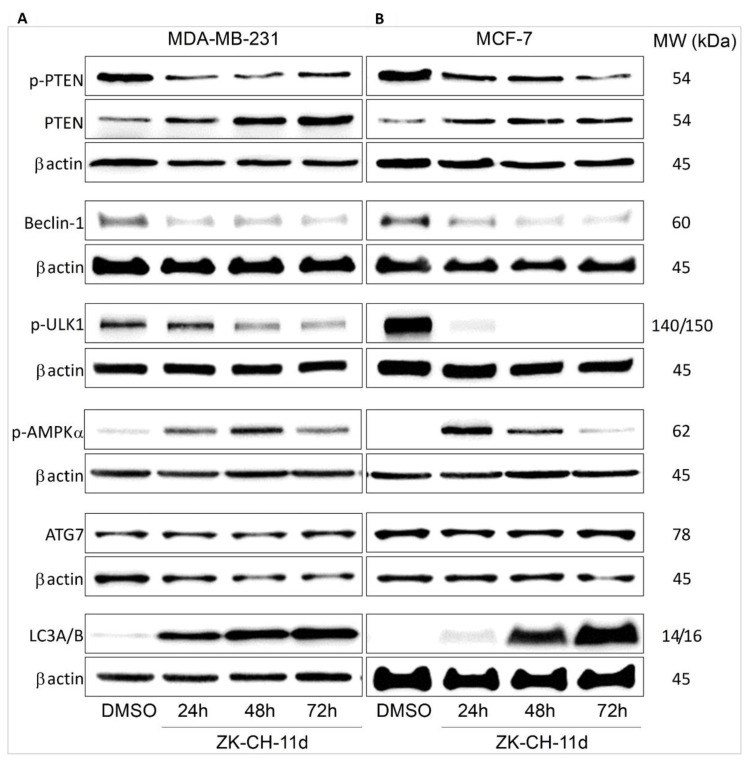
Western blot analysis of autophagy and autophagy-related proteins in MDA-MB-231 (**A**) and MCF-7 (**B**) cells after treatment with chalcone ZK-CH-11d for 24, 48, and 72 h of treatment.

**Figure 17 pharmaceutics-14-00503-f017:**
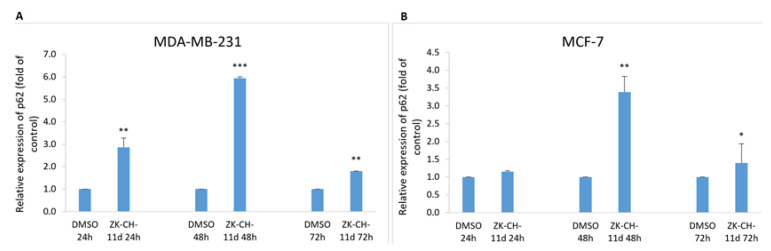
Relative expression of p62 in MDA-MB-231 (**A**) and MCF-7 (**B**) cells after chalcone ZK-CH-11d treatment for 24, 48, and 72 h. Statistical significance: * *p* < 0.05, ** *p* < 0.01, *** *p* < 0.001 vs. vehicle (DMSO).

**Figure 18 pharmaceutics-14-00503-f018:**
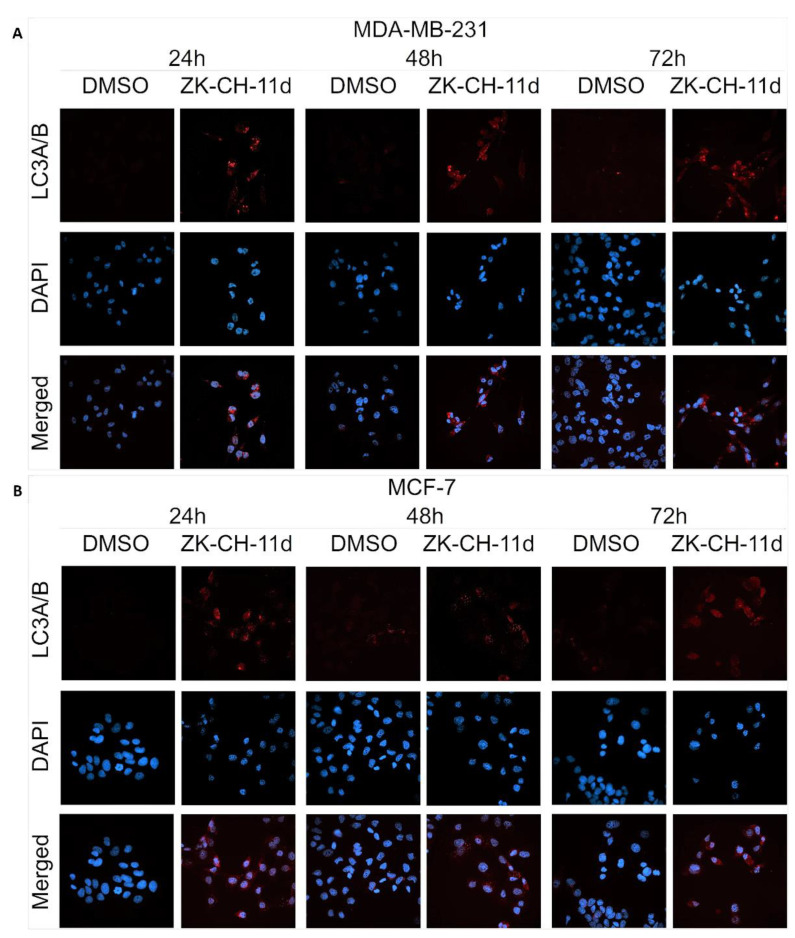
Immunofluorescence detection of LC3A/B in MDA-MB-231 (**A**) and MCF-7 (**B**) cells treated with chalcone ZK-CH-11d. Red represents the LC3A/B protein and blue represents DAPI-stained nuclei. The figure is a representative average of three independent experiments. Magnification: 60×.

**Figure 19 pharmaceutics-14-00503-f019:**
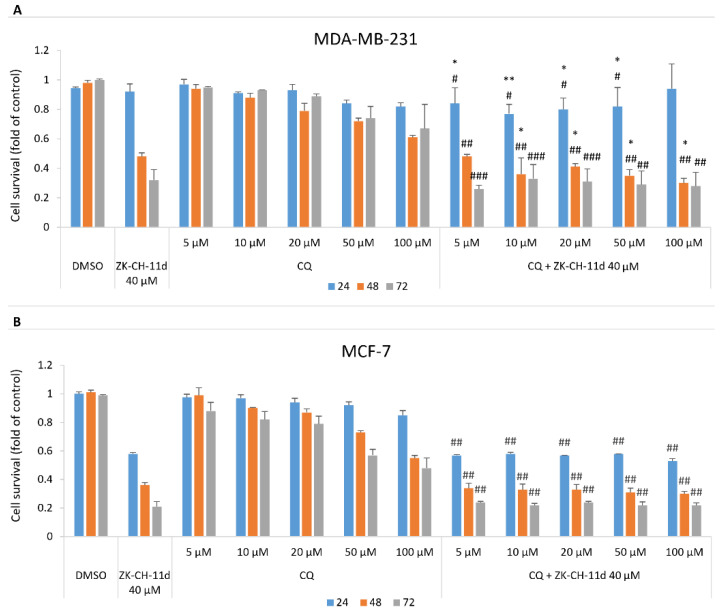
Effect of ZK-CH-11d and chloroquine on the proliferation of MDA-MB-231 (**A**) and MCF-7 (**B**) cells after 24, 48, and 72 h. The data are the results of three independent experiments. Values represent the mean ± standard deviation. Statistical significance: * *p* < 0.05, ** *p* < 0.01, vs. ZK-CH-11d; # *p* < 0.05, ## *p* < 0.01, ### *p* < 0.001 vs. chloroquine.

**Table 1 pharmaceutics-14-00503-t001:** List of flow cytometric antibodies and stating solutions.

Analysis	Staining Solution	Manufacturer
Caspase 3	Cleaved Caspase-3 (Asp175) (5A1E) Rabbit mAb (PE Conjugate), 1:300	Cell Signaling Technology^®^, Danvers, MA, USA
Caspase 7	Cleaved Caspase-7 (Asp198) (D6H1) Rabbit mAb (PE Conjugate), 1:300	
PARP	Cleaved-PARP (Asp214) XP^®^ Rabbit mAb (PE Conjugate), 1:300	
p53	p53 (1C12) Mouse mAb (Alexa Fluor^®^ 488 Conjugate), 1:300	
Phospho-p53	Phospho-p53 (Ser15) Mouse mAb (PE Conjugate), 1:300	
Phospho-ATM	Anti-pATM, PE conjugated antibody, 1:200	Millipore Corporation, Temecula, CA, USA
Phospho-SMC1	Anti-pSMC1, Alexa Fluor 488 Antibody, 1:200	
Cytochrome *c*	Cytochrome *c* Antibody (6H2) FITC Conjugate, 1:200	Invitrogen, Carlsbad, CA, USA
p62	p62Anti-SQSTM1/p62 antibody, Alexa Fluor 488 antibody, 1:500	Abcam, Cambridge, United Kingdom
8-oxoguanin	Anti-oxoguanine 8 antibody, 1:200	
MMP	TMRE (Tetramethylrhodamine ethyl ester perchlorate), final concentration 0.1 µmol/L	Sigma-Aldrich, St. Louis, MO, USA
Secondary antibody	Goat anti-mouse IgG (H + L) secondary antibody, Alexa Fluor 488, 1:300	Thermo Scientific, Rockford, IL, USA

**Table 2 pharmaceutics-14-00503-t002:** List of Western blot antibodies.

Primary Antibody	Mr (kDa)	Origin	Manufacturer
α tubulin	55	Rabbit	Santa Cruz Biotechnology, Inc. (Dallas, TX, USA)
α 1c tubulin	50	Mouse	
β tubulin	55	Rabbit	
Caspase 3	11/17/20/32	Mouse	
Caspase 7	10/20/28/38	Mouse	
Bax	23	Mouse	
Bcl-xL	30	Rabbit	Cell Signaling Technology^®^, Danvers, MA, USA
PARP	116/89	Rabbit	
Phospho-Survivin	16/18	Rabbit	
p21	21	Rabbit	
p27	27	Rabbit	
Phospho-Rb	110	Rabbit	
Phospho-Wee1	95	Rabbit	
Phospho-Cyclin B1	55	Rabbit	
Cyclin D1	36	Rabbit	
Phospho-Cyclin D1	36	Rabbit	
Phospho-cdc2	34	Rabbit	
Phospho-Chk1	56	Rabbit	
PI3 Kinase p85	85	Rabbit	
Phospho-PI3 Kinase p85/p55	60/85	Rabbit	
Akt	60	Rabbit	
Phospho-Akt	60	Rabbit	
mTOR	289	Rabbit	
Phospho-mTOR	289	Rabbit	
PTEN	54	Rabbit	
Phospho-PTEN	54	Rabbit	
Beclin-1	60	Rabbit	
Phospho-ULK1	140–150	Rabbit	
Phospho-AMPK α	62	Rabbit	
Atg7	78	Rabbit	
LC3A/B	14/16	Rabbit	
Erk1/2	42/44	Rabbit	
Phospho-Erk1/2	42/44	Mouse	
p38 MAPK	43	Rabbit	
Phospho-p38 MAPK	43	Rabbit	
Phospho-Histone HA2.X	15	Rabbit	
β actin	45	Mouse	
**Secondary Antibody**	**Mr (kDa)**	**Origin**	**Manufacturer**
Anti-rabbit IgG HRP	-	Goat	Cell Signaling Technology, Danvers, MA, USA
Anti-mouse IgG HRP	-	Goat	

**Table 3 pharmaceutics-14-00503-t003:** IC_50_ (μmol/L) of ZK-CH-11d on breast cancer cell line MDA-MB-231 and MCF-7 and non-cancer cell line MCF-10A.

Compound	Cell Line		
	MDA-MB-231	MCF-7	MCF-10A
ZK-CH-11d	34.03 ± 3.04	37.32 ± 1.51	>100

## Data Availability

The data presented in this study are available in Appendix A or can be provided by the authors.

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
