# Peer review of "Programmed Cell Death Alterations Mediated by Synthetic Indole Chalcone Resulted in Cell Cycle Arrest, DNA Damage, Apoptosis and Signaling Pathway Modulations in Breast Cancer Model"

_pharmaceutics, 2022, doi:10.3390/pharmaceutics14030503_

Round 1
Reviewer 1 Report
Manuscript seems to be interested and valuable, yet it needs to be well organized. In the manuscript, there are a lot of mistakes: the figure number are unarranged, the chart axis descriptions are incorrect, the style of the charts is patchy, etc. These mistakes hamper the follow of the Authors idea and the interpretation of obtained results. The Reader may make a presumption that many presented results are confused or they were jumbled. The abbreviation should be explained, especially in legend description. The text needs to be well-checked and English correction is needed. These errors prevent reliable and substantive verification of the work. However, I believe that it is only wrong version submitted to the journal. The complex analysis of apoptosis and autophagy markers and cell cycle analysis were made. The presented results may have a high impact on development of experimental therapy of breast cancer cells.
Briefly speaking, I deem the topic of the manuscript noteworthy as well as attractive, yet it should be amended.
Author Response
We thanks reviewer for the opportunity to improve our manuscript.
Manuscript seems to be interested and valuable, yet it needs to be well organized.
In the manuscript, there are a lot of mistakes: the figure number are unarranged, the chart axis descriptions are incorrect, the style of the charts is patchy, etc. These mistakes hamper the follow of the Authors idea and the interpretation of obtained results. The Reader may make a presumption that many presented results are confused or they were jumbled.
We thanks reviewer for valuable comments. We checked again figures numbering, corrected and unify chart axis.
The abbreviation should be explained, especially in legend description.
Accepted, the abbreviation were explained when necessarily.
The text needs to be well-checked and English correction is needed. These errors prevent reliable and substantive verification of the work.
Thank you for your opinion, the native speaker checked manuscript.
However, I believe that it is only wrong version submitted to the journal. The complex analysis of apoptosis and autophagy markers and cell cycle analysis were made. The presented results may have a high impact on development of experimental therapy of breast cancer cells.
Briefly speaking, I deem the topic of the manuscript noteworthy as well as attractive, yet it should be amended.
Reviewer 2 Report
The manuscript pharmaceutics-1551313 ”Synthetic chalcone as an antiproliferative agent induces cell cycle arrest, apoptosis, DNA damage, autophagy and modulates selected signalling pathways in breast cancer cells” by Radka Michalkova et al presents some studies about the potential anticancer activity of one synthetic chalcone (ZK-CH-11d). The article is well-written and presents interest for the Pharmaceutics journal.
As suggestions/comments:
- The reference for the test compound should be mentioned in section 2.1
- Please check the information presented in rows 232-234, or specify the non-toxic DMSO concentration
-
How can you explain the low IC50 values ​​for the compound tested for cancer cell lines and high for non-cancer line (rows 229-231)?
- Why did you choose a concentration of 40 μmol/L compound?
- Please review the writing of the bibliography (uniform writing of the bibliographic indexes and in accordance with the requirements of the journal)
Author Response
We thanks reviewer for the opportunity to improve our manuscript.
The manuscript pharmaceutics-1551313 ”Synthetic chalcone as an antiproliferative agent induces cell cycle arrest, apoptosis, DNA damage, autophagy and modulates selected signalling pathways in breast cancer cells” by Radka Michalkova et al presents some studies about the potential anticancer activity of one synthetic chalcone (ZK-CH-11d). The article is well-written and presents interest for the Pharmaceutics journal.
As suggestions/comments:
- The reference for the test compound should be mentioned in section 2.1
Accepted. The mentioned compound characteristics and synthesis will be published soon. Therefore we have not yet references, however we added additional supported data to suplement file.
2. Please check the information presented in rows 232-234, or specify the non- toxic DMSO concentration
Corrected, non-toxic DMSO concentration value is mentioned in the text now.
3. How can you explain the low IC50 values ​​for the compound tested for cancer cell lines and high for non-cancer line (rows 229-231)?
Thank you for this comment. The metabolism of cancer cells can differ markedly from healthy cells. For example, cancer cells consume far more glucose to generate energy and to produce materials that support cell division. The faster metabolisms, differences in transport systems as ABC transporters that are connected with membrane transport of nutrients and also drugs influence significantly the effect of tested substances. Normal cells can eliminate tested compound by efflux more efficient. This fact is also in accordance with that potential anti-cancer drugs should be less toxic to normal cells. In our case we confirmed that tested chalcone need much more higher concentration to kill 50% of normal cells.
4.Why did you choose a concentration of 40 μmol/L compound?
We thanks reviewer for the comment. We adjusted concentrations a little higher compared to IC50 values. Metabolic screening tests are only approximate (as seen on SD values) and first experiments revealed that IC50 should be a little higher. We increased final conc. from 34 resp. 37 to 40 μmol/L.
5. Please review the writing of the bibliography (uniform writing of the bibliographic indexes and in accordance with the requirements of the journal)
Corrected, we used Endnote software but some differences occurred in database.
Round 2
Reviewer 1 Report
This version of the manuscript is very good and interesting. I have a few comments, namely:
1. specify which IC50 refers to which cell line
2. add densitometric analysis under each blot - this will make interpretation of results easier
3. arrange the figures according to the order of their description - Figures 9-11.
4. line 590 - specify why in Figure 16 AMPK in MCF-7 cells increases relative to control but decreases relative to incubation time - it needs specify
5. line 661 - what does (Peter, 2002) mean?
6. After careful analysis of the paper I suggest specifying the manuscript title, especially as the authors wrote „Our results indicate that antiproliferative effect of chalcone ZK-CH-11d is associated with induction of apoptosis, cell cycle arrest in G2/M phase, DNA damage and inhibition of PI3K/Akt/mTOR pathway and autophagy.” (line 68-71), but the current title of the paper does not fully convey the main message of the article, especially in terms of the autophagy proces. It may confuse the reader.
7. There are so many acronyms used in this work that it would be convenient to include a section on abbreviations
Author Response
Hallo, we thanks reviewer for additional comments. We adress all below.
This version of the manuscript is very good and interesting. I have a few comments, namely:
- specify which IC50 refers to which cell line
The IC50s are mentioned in Table 3 and in the text section 3.1 and 3.2 where MTT and BrdU results are described. For further experiment, we decided to use a concentration of 40 μmol/L for both cell lines as is also written.
- add densitometric analysis under each blot - this will make interpretation of results easier
We agree with this argument and we understand it but the realisation would be difficult because we should add additional 40 graphs to final Figures. If we add densitometry to the main figures, the figures will be too big in format A4 even if we decrease size of graphs (in some cases even unreadable). Moreover, manuscript is huge now and will be much more longer with new figures. Therefore, we realised published densitometry as Supplement.
- arrange the figures according to the order of their description - Figures 9-11.
Accepted, the figures were re-arranged according to more logic order
- line 590 - specify why in Figure 16 AMPK in MCF-7 cells increases relative to control but decreases relative to incubation time - it needs specify
Phosphorylation of ULK1 at Ser757 is thought to inhibit its important function in initiating autophagy (via phospho-mTOR). In contrast, phosphorylation of ULK1 at Ser113 and Ser777 is mediated by phospho-AMPK alpha, indicating ULK1 activation. We analyzed changes in the phosphorylation status of phospho-ULK1 on Ser757 to monitor the direct effect of mTOR on ULK1 inhibition. Therefore, we suggest that the time-dependent decrease in phosho-AMPK alpha is related to its effect on ULK1 at Ser113 and Ser777 and thus its activation. We believe that significant induction of apoptosis as the predominant mechanism may also have an effect on the decrease in this kinase level.
- line 661 - what does (Peter, 2002) mean?
Sorry for the mistake, the citation now is in proper format as citation number 44
- After careful analysis of the paper I suggest specifying the manuscript title, especially as the authors wrote „Our results indicate that antiproliferative effect of chalcone ZK-CH-11d is associated with induction of apoptosis, cell cycle arrest in G2/M phase, DNA damage and inhibition of PI3K/Akt/mTOR pathway and autophagy.” (line 68-71), but the current title of the paper does not fully convey the main message of the article, especially in terms of the autophagy proces. It may confuse the reader.
Accepted
New title:
Programmed cell death alterations mediated by synthetic indole chalcone resulted in cell cycle arrest, DNA damage, apoptosis and signalling pathway modulations in breast cancer model.
- There are so many acronyms used in this work that it would be convenient to include a section on abbreviations
Thank you for this comment. All abbrevations are described in text in the place when 1st time occurred. Moreover, we also added short Abbrevation list after Conlusion section.